# Superordinate Categorization Based on the Perceptual Organization of Parts

**DOI:** 10.3390/brainsci12050667

**Published:** 2022-05-20

**Authors:** Henning Tiedemann, Filipp Schmidt, Roland W. Fleming

**Affiliations:** 1Department of Experimental Psychology, Justus Liebig University Giessen, 35394 Giessen, Germany; filipp.schmidt@psychol.uni-giessen.de (F.S.); roland.w.fleming@psychol.uni-giessen.de (R.W.F.); 2Center for Mind, Brain and Behavior (CMBB), University of Marburg and Justus Liebig University Giessen, 35037 Marburg, Germany

**Keywords:** visual perception, objects, shape, features, visual cognition

## Abstract

Plants and animals are among the most behaviorally significant superordinate categories for humans. Visually assigning objects to such high-level classes is challenging because highly distinct items must be grouped together (e.g., chimpanzees and geckos) while more similar items must sometimes be separated (e.g., stick insects and twigs). As both animals and plants typically possess complex multi-limbed shapes, the perceptual organization of shape into parts likely plays a crucial rule in identifying them. Here, we identify a number of distinctive growth characteristics that affect the spatial arrangement and properties of limbs, yielding useful cues for differentiating plants from animals. We developed a novel algorithm based on shape skeletons to create many novel object pairs that differ in their part structure but are otherwise very similar. We found that particular part organizations cause stimuli to look systematically more like plants or animals. We then generated other 110 sequences of shapes morphing from animal- to plant-like appearance by modifying three aspects of part structure: sprouting parts, curvedness of parts, and symmetry of part pairs. We found that all three parameters correlated strongly with human animal/plant judgments. Together our findings suggest that subtle changes in the properties and organization of parts can provide powerful cues in superordinate categorization.

## 1. Introduction

Visually differentiating between superordinate classes, e.g., plants and animals, is a vital skill with direct consequences for our behavior. For example, many animals can move at speeds that demand immediate action—such as fleeing or attacking—while plants do not. However, the visual distinction between these classes is not always easy. Even though objects within a superordinate class typically share particular characteristics (e.g., visible motion), they might still look very different. For example, elephants and insects both belong to the animal class, but are very different in their visual appearance. In contrast, insects and twigs can share many visual features, such as thin and spindly limbs, but belong to different superordinate classes. Consequently, we need to be able to quickly analyze and interpret sometimes subtle shape differences to achieve correct categorizations and adjust our behavior accordingly.

Previous research identified some of the visual aspects distinguishing superordinate classes like animate and inanimate objects [1,2,3] as well as aspects of the neural processing that drives categorization [4,5,6]. Specifically, these studies focused on diagnostic local features such as eyes and mouth (e.g., [1]), individual parts such as tails [7], global features such as canonical animal postures [8], or overall curvature [9,10,11]. However, all of these studies are virtually agnostic with respect to the role of perceptual organization of object shape, that is, the spatial arrangement and relations between the different parts of an object. Hereinafter, we use the term “part structure” to refer to the visual segmentation of an object into its parts (e.g., [12,13]) as well as to the analysis of relational information—for example, to which other parts a given part is connected (e.g., [14]), whether parts form a symmetrical pair (e.g., [15]), or other non-accidental configurations like collinearity of parts [16,17].

There are several reasons why part structure might contribute to superordinate object categorization. First, natural growth processes tend to follow systematic rules [18], resulting in highly regular growth patterns. This constrains the perceptual organization of object parts in otherwise visually highly varied categories (Figure 1). For example, the presence or absence of symmetrical parts is a distinctive quality of part structure. Animals such as insects, elephants and vultures could hardly be more visually distinct, yet all of them are comprised of a main body with prominent pairs of symmetrical limbs, most often arms and legs. Indeed, biologically, the vast majority of animals follow a bilaterally symmetric body plan [19] supporting the notion that humans have evolved (or learned) to expect this part structure in animals. In line with this, previous studies have found that symmetry is indeed a strong animal-cue [3]. In contrast, plants such as oak trees or ferns typically grow parts radially in a Fibonacci sequence [20], resulting in a roughly alternating left/right placement of parts when viewed from the side. Experiments using abstract shapes have shown that regularities in part structure can, in certain situations, drive categorization [21], supporting the notion that these ubiquitous natural regularities play a role as well.

Second, part structure is a useful visual feature because it is independent of temporary limb articulation. For example, an ape rarely positions its limbs in perfect mirror symmetry formation. Instead, the limbs are configured according to the task currently being performed. However, by attending to the part structure, that is, to where the limbs are attached to the ape’s main body, an observer can abstract away from their current shape and positioning. Although limb pose can be highly flexible, an animal’s part structure rarely changes with motion, so it is an immutable feature of its growth history and class.

Third, animals and plants can be differentiated perceptually by the presence of higher-order parts. Animals mainly consist of a main body with prominent first-order parts (limbs that grow directly from the main body). In contrast, plants are often composed of a hierarchy including many higher-order parts, i.e., parts that do not connect directly to the main body (red in Figure 1) but rather derive from a first-order or a higher-order part (orange to green parts in Figure 1). These higher-order parts of plants often sprout off their parents, resulting in sharp angles that make them more salient and visually distinct. In contrast, if animals possess higher-order parts at all—for example, fingers or claws—they often grow off the parent part’s end, and the resulting obtuse angles make them look more like a continuation of the parent than distinct parts on their own. As a result of these typical differences, symmetry of part pairs and sprouting higher-order parts are potentially useful cues for distinguishing plants from animals.

Previous findings suggest that part structure is an important feature for the speed of object identification [22] and for discriminating between abstract shapes [23,24]. Other findings indicate that part segmentation is available early in visual processing [25,26]. Together, this suggests that part structure might be a powerful cue for superordinate categorization of objects. Here, we ask to what extent humans make use of such cues.

We can also use differences in part structure to map out the superordinate category boundary along the spectrum of shapes spanning plants and animals. Previous studies did this by generating shapes along a continuum of superordinate classes (e.g., animals and leaves [27]) using a global shape-morphing algorithm to move between contours of different classes (e.g., [28]). This method of global shape-morphing, however, can often result in shapes that do not belong clearly to either of the superordinate classes and, moreover, does not permit testing the role of specific individual cues in participants’ judgments easily. Here, we used our observations about systematic differences in part structures between animals and plants to morph continuously between classes by parametrically varying these cues. We focused on novel, generated objects to provide greater control over the parameters than is possible through photographs or silhouettes of natural shapes and to help reduce the influence of semantic knowledge and ground-truth category membership. We reasoned that while objects are generated to possess more or fewer features of a particular class, their novelty and the continuous variation of parameters ensures that the superordinate categorization task is nontrivial.

## 2. Experiment 1: Differences in Part Symmetry and Curvedness

To investigate the role of part symmetry as a superordinate categorization cue, we sought to generate pairs of shapes that differed only in that aspect—with one having symmetrical pairs of parts and the other having the same parts slightly displaced with respect to each other. Previous studies also found curvature to be a distinguishing factor between animals and plants (e.g., [9]); however, they analyzed and manipulated curvature at the image level. In contrast, we decided to investigate the role of curvature by creating pairs of shapes that only differed in the articulation of parts—namely, in straighter or curved limbs. To achieve these aims, we developed a generative algorithm based on shape skeletons [14,29,30] that allowed us to vary shape parameters such as the number of skeletal limbs (parts) and the width and length of each part (Figure 2). This algorithm was used to create all stimuli in this study.

### 2.1. Participants

The participants (*n* = 13) were recruited through the university mailing list. All the participants gave informed consent before the experiments in accordance with the Declaration of Helsinki. The procedures were approved by the Local Ethics Committee of the Department of Psychology and Sports Sciences of Justus Liebig University Giessen.

### 2.2. Materials and Methods

For Experiment 1, we created pairs of shapes with the simultaneous aims of maximizing visual similarity while varying symmetry of part pairs or curvedness of parts. Therefore, each shape pair consisted of the same main body, which was a simple elongated shape in upright orientation, with the same number of limbs growing from the main body of both shapes.

For the symmetry condition, one limb of each part pair in the second shape was displaced (top row of Figure 3)—that is, moved randomly up or down the main body by either 2.5%, 5.0%, 7.5%, 10.0%, or 12.5% of the main body’s perimeter—creating pairs with subtle to very large asymmetries.

For the curvedness condition, we firstly created the limbs of the first shape with high curvedness (the bottom row of Figure 3), with angles between joints ranging from 50 to 100 degrees. For the second shape, these limbs were “straightened out” by moving the individual joints so that the resulting angle between all the joints was reduced by one of five factors (ranging from 0.05 to 0.25 in steps of 0.05). For example, multiplication using a factor of 0.25 with a limb’s joints with an angle of 100° resulted in a limb with joints with an angle of 25° while keeping the distances between them the same. As a result, the parts of the second shape varied between somewhat curved to being almost completely straight.

For both conditions, the number of limbs attached to the main body varied from two to five. Twenty shape pairs were created for each number of limbs (4) and parameter values (5) resulting in 20 × 4 × 5 = 400 stimuli per condition. In the 2-AFC task, the participants were presented with each shape pair and instructed to identify which of the two shapes belongs to the category of plants—and implicitly that the other belongs to the category of animals—with a press of a button. Thus, the same shape could never be plant-like and animal-like at the same time. They were further instructed that there were no right or wrong answers and, if unsure, to pick the response that fit the respective categories best. No further elaborations as to the research question were made. In spite of the shapes’ relatively abstract visual appearance, the participants reported no problem in performing the task. The shape pairs were shown side by side in randomized order and with randomized left/right placement.

### 2.3. Results

Bar plots in Figure 3 show the results for both symmetry of part pairs (top) and curvedness (bottom) on a per-participant basis. The shapes with asymmetrical part pairs were judged to be more plant-like than their counterparts by all but one participant, resulting in a strong overall preference in 91% of the trials (one-sided binomial test: 50%; N = 5200, i.e., number of judgements; K (correct responses) = 4735, *p* < 0.001), showing that this aspect of part structure is indeed a strong differentiating cue between plants and animals. Curvedness also had a significant, although smaller, effect on categorization with less curved shapes seen as more plant-like in 75% of the trials (one-sided binomial test: 50%; N = 5200, i.e., number of judgements; K (correct responses) = 3879, *p* < 0.001)—with the exception of two participants who displayed the opposite pattern, with more curved shapes appearing more plant-like. This suggests that part structure is an effective cue for superordinate categorization, just like part articulation.

As described above, asymmetry and straightness varied from subtle to strong in five steps to test if larger differences in these two aspects would in turn lead to stronger categorization effects. A one-sided ANOVA for the different levels of asymmetry was significant (F(4,395) = 3.49, *p* < 0.01). Performing Tukey’s post-hoc tests showed that the smallest asymmetry (displacement of 2.5% of the main body perimeter) was significantly different from both the second-largest and the largest displacements at *p* < 0.05 (10% and 12.5% displacement, respectively). Note that these differences are, however, relatively small in absolute numbers, with the smallest asymmetry seen as plant-like 88% of the time, compared to 93% and 92% for the two largest asymmetries. The ANOVA for the different levels of curvedness was not significant (F4,395 = 0.76, *p* = 0.55).

## 3. Experiment 2: Differences in Part Hierarchy between Animals and Plants

To investigate the role of part hierarchy in superordinate categorization, we generated shape pairs that differed in whether they exhibited higher-order limbs (i.e., sprouting) but were otherwise similar in appearance.

### 3.1. Participants

The participants (*n* = 15) were recruited through the university mailing list. All the participants gave informed consent before the experiments in accordance with the Declaration of Helsinki. Procedures were approved by the Local Ethics Committee of the Department of Psychology and Sports Sciences of Justus Liebig University Giessen.

### 3.2. Materials and Methods

Equivalently to Experiment 1, we created shape pairs only differing in the presence of second-order limbs. In each pair, the first shape consisted of the main body and two sets of symmetrical first-order limbs, emulating the look of an insect-like animal. The second shape consisted of the same main body and first-order limbs rearranged to imitate the typical part structure of a tree-like plant (by “transplanting” one set of first-order limbs onto another set, Figure 4a). As Experiment 1 showed that symmetry of part pairs and curvedness of limbs are strong categorization cues which might modulate potential effects of higher-order limbs, we created four groups of stimulus pairs from all the possible combinations of symmetrical/asymmetrical and curved/straight limbs (Figure 4a). Thus, we were able to test the effect of sprouting limbs on shapes that we expected to appear more plant-like (straight and asymmetrical limbs), animal-like (curved and symmetrical limbs), or in-between (straight and symmetrical as well as curved and asymmetrical limbs). For each of these four stimulus types, we created 125 shape pairs, resulting in 500 trials with no repetitions.

The participants were instructed to identify one shape that was more plant-like versus another shape that was more animal-like, meaning one shape could not be both more animal- and plant-like, as in Experiment 1. Apart from the stimuli, the experiment was identical to Experiment 1: the shape pairs were shown side by side in randomized order and with randomized left/right placement. The participants would then choose the more plant-like object with a press of a button (which also defined the other shape as more animal-like). This way we could test to what extent the presence of second-order limbs served as a cue to superordinate categorization of plants and animals.

### 3.3. Results

Figure 4b shows the results for Experiment 2 on a per-participant basis. The shapes with sprouting limbs (i.e., with second-order limbs) were consistently judged to be more plant-like than their counterparts and, conversely, the shapes without sprouting were judged to be more animal-like (one-sided binomial test: 50%; N = 7500; K = 6205, *p* < 0.001). This illustrates the effectiveness of sprouting limbs in making a shape seem distinctively more plant- and less animal-like.

With respect to the additional parameters of symmetry and curvedness, we found no significant difference between the four stimulus types (ANOVA: F3,496 = 1.42, *p* = 0.24). This suggests that sprouting parts are the main driving force in this stimulus set independent of the other available cues.

## 4. Experiment 3: Morphing from Animals to Plants with Changes in Part Structure

After having established that features of part structure are effective cues for differentiating between animals and plants, we set out to investigate whether continuous changes in these features across the spectrum would result in corresponding changes in categorical decisions. Previous studies morphed between two end-states of different categories (e.g., animals and leaves [27]). However, by using global shape-morphing algorithms based on contours, in-between morphs did not always look like real objects belonging to any specific category. In contrast, we manipulated objects in a part-based fashion so that the number of parts as well as the parts themselves (apart from articulation) stayed the same throughout the morphing sequence. This way we could test whether differences in part structure alone are sufficient to define perceived category membership, without additional cues like texture or part identity (e.g., leaves or wings). Specifically, we asked whether the “animal-ness” of an object decreases continuously with the increase in the corresponding plant-like changes in part structure up to an ambiguous midpoint after which the object appears more like a plant. Alternatively, it would be possible that the chosen parameters were not sufficient to induce gradual changes in category membership. In that case, all shapes along the morphing spectrum might look either animal-like or plant-like (e.g., when changes in part curvedness would be interpreted as the same object with a different articulation of its limbs rather than another object from a different class). The often-subtle differences in part structure between morphing steps (in contrast to more blatant differences like the number of parts) allowed us to test the type of fine-grained categorical decisions that are often necessary in the real world, for example, when differentiating twigs from insects. Given that single parameters tend to explain little variance in the distinctions between superordinate classes [2,3], we decided to vary all three parameters from Experiments 1 and 2 together (symmetry of part pairs, curvedness of parts, and sprouting parts).

### 4.1. Participants

The participants (*n* = 14) were recruited through the university mailing list. All the participants gave informed consent before the experiments in accordance with the Declaration of Helsinki. The procedures were approved by the Local Ethics Committee of the Department of Psychology and Sports Sciences of Justus Liebig University Giessen.

### 4.2. Materials and Methods

We morphed along three parameters at the same time: (1) the symmetry of part pairs, varying between symmetrical and asymmetrical; (2) the curvedness of parts, varying between curved and almost straight; (3) the organization of parts, varying in two different ways (Figure 5). The first type of stimuli featured a more plant-like form of sprouting (growing sprouting condition): sprouting limbs grew from random points on their parent limb and were articulated independently, without making a symmetrical pair with the sprout on the corresponding mirrored parent part. Furthermore, we varied the size of the sprouts, with second-order limbs growing from non-existent to relatively large. The second type of stimuli featured sprouting limbs with more animal-like characteristics (symmetrical sprouting condition): second-order limbs formed symmetrical pairs (if the parent limb was also part of a symmetrical pair) and were of the same size throughout the morphing sequence—emulating real-world second-order limbs of animals like claws or thumbs. In both (growing sprouting, symmetrical sprouting) conditions, the sprouting limbs changed in the same way as the first-order limbs did along the morphing sequence—they became straighter and more asymmetrical. Along with these parameters, shapes varied in the number of their first-order limbs (from two to six; held constant along the morphing sequence).

With these parameters, we created 110 sequences of five morph levels from animal- to plant-like, resulting in 550 shapes in total. Each shape consisted of a vertically oriented main body, with a different number of first-order limbs (ranging from two to six) that formed symmetrical pairs. With odd numbers of limbs, the additional limb would not have a symmetrical counterpart.

All the 550 shapes were shown in randomized order one by one so that the morphing sequences were not apparent to the viewer. The participants were asked to rate each shape on a continuous scale from animal- to plant-like by moving a slider, with the midpoint representing a shape that is equally animal- and plant-like.

The different types of stimuli allowed us to investigate two questions. First, can we generalize the previous findings of Wilder et al. (2011) by showing that a larger number of limbs not only makes an object look more like a leaf, but also generally more like a plant? If that were the case, we should find that shapes with more first-order limbs are judged to be more plant-like compared to shapes with fewer first-order limbs. Second, are shapes with an odd number of first-order limbs, and therefore no bilateral symmetry, more biased towards the plant side than shapes with an even number of limbs?

### 4.3. Results and Discussion

The participants’ responses were closely related to variations in our parameter space, in which symmetry, curvedness, and sprouting were manipulated simultaneously. Shapes with more animal-like parts (more curved, symmetrical, and with fewer prominent second-order limbs) were perceived as more animal-like and vice versa for more plant-like parts (Figure 6). To summarize the sigmoidal trends in the data, we fit psychometric functions to all the responses using psignifit [31]. Overall, the responses were biased towards the plant end of the spectrum (first panel of Figure 6) with the average categorization value of 0.59 (most animal-like = 0, most plant-like = 1)—documented by a one-sample *t*-test comparing all the responses with the midpoint of 0.5 (t(7699) = 28.47, *p* < 0.001). This bias might be explained by the existence of (albeit small) sprouting limbs in all the conditions, except for the most animal-like stimulus type in the “growing sprouting” condition (meaning that 90% of the stimuli had some form of second-order limbs). Furthermore, only the most animal-like step had complete symmetry among the limb pairs, with the other four steps introducing asymmetry, a potent plant cue, potentially adding to the plant bias. Moreover, it is also worth noting that we did not systematically manipulate the local symmetry of second-order limbs meaning that they may have more closely resembled plant branches than animal limbs.

Different sprouting styles (second panel of Figure 6) elicited highly similar responses, showing that they had roughly the same impact on categorization. An ANOVA yielded no significant difference between the two sprouting styles (F(1,548) = 0.01, *p* = 0.92). To establish whether we had evidence for no difference between the sprouting styles, we calculated a scaled JZS Bayes factor using a Jeffrey–Zellner–Siow prior (Cauchy distribution on effect size) with a default scale factor of 0.707 [32], resulting in a BF10 of 0.03. This measure describes the probability of the data given H1 relative to H0, where a BF10 < 0.1 can be considered “strong evidence” for H0 [32]. This suggests that, across the complete morphing sequence, it did not matter whether sprouting limbs grew from random points and varied in size (growing sprouting) or whether they formed symmetrical pairs that did not change in size (symmetrical sprouting).

Finally, increasing numbers of limbs attached to the main body (third and fourth panels in Figure 6) increased the slope of the fit function, indicating a stronger distinction between animal- and plant-like shapes. While the number of parts did not affect the plant side of the morphing spectrum much, it strongly affected the judgement of animal-like shapes. This is corroborated by separate ANOVAs for each morphing step, showing significant differences within steps 1 (most animal-like) to 3 (midpoint), but not within steps 4 and 5 (most plant-like; Bonferroni-adjusted *p*-value of 0.01 for five tests). Consequently, the number of limbs only affected judgement of ambiguous and animal-like shapes—with animal-like shapes with fewer limbs judged as much more plant-like compared to animal-like shapes with many limbs. Furthermore, an odd number of limbs created an overall more plant-like impression along the morphing sequence (fourth panel in Figure 6), with a smaller slope compared to shapes with an even number of limbs (one limb either more or less), presumably because animals canonically have an even number of limbs.

## 5. Discussion

Visually distinguishing between superordinate classes is a vital yet challenging skill. Different members of the same superordinate class may share certain features, but also differ from one another in other highly salient ways. Somehow, the visual system must determine which feature differences are relevant for superordinate classification and which otherwise significant differences in appearance ought to be ignored. How humans achieve this remains unclear. Most previous studies investigated local object features such as eyes and mouths (e.g., [1]), individual parts such as tails [7], global features such as canonical animal posture [8], or overall curvature [9,10,11] without considering the perceptual organization of objects. However, since organic objects grow in specific ways according to particular laws (e.g., [18,20]), the resulting visual part structure is highly constrained by biological class, therefore specifying a potentially powerful cue for superordinate categorization.

Based on this observation, we conducted two categorization experiments with pairs of shapes that were identical except for the symmetry of part pairs, the curvedness of parts, or the presence of second-order limbs (Experiments 1 and 2). We found that both symmetry and curvedness were effective cues for differentiating between animals and plants. The presence of second-order limbs was also used to categorize novel objects as plants or animals. Together, these findings suggest that the perceptual organization of objects—the spatial arrangement and relations between their parts—helps us to distinguish between superordinate classes of animals and plants, and potentially other superordinate categorizations.

In the final experiment, we used these newly identified categorization cues to create sequences of shapes morphing from animal- to plant-like in a continuous, part-based fashion. Since superordinate categorization tends to be affected by multiple factors at once [2,3], all three features varied simultaneously. We also included two different styles of sprouting, one closer to the biological reality of plants (growing sprouting condition) and the other emulating animal-like second-order limbs (symmetrical sprouting). We found that the three categorization cues predicted category membership very well even if the two sprouting styles did not create appreciable differences in categorization. Additionally, the more parts a shape had, the stronger the perceived difference across the morphing spectrum, especially for shapes with animal characteristics: the shapes with fewer than four parts were seen as more plant-like across the spectrum compared to shapes with four or five limbs. It is interesting to speculate whether this might have been because canonically mammals possess four limbs, whereas plants deviate more often from this number. Finally, shapes with an odd number of first-order limbs were more biased towards the plant end of the spectrum regardless of curvedness or symmetry of part pairs compared to even-numbered counterparts—showing a further effect of structural symmetry on categorization (i.e., the symmetry of the points of attachment along the length of the body). Overall, our findings show that even small differences in part structure can change the superordinate categorization of objects as opposed to more blatant changes like adding new parts [27] or distinctive features [1,7], emphasizing just how subtle decisive category features can be.

Given the speed of visual part segmentation [25,26] and categorization [1] in humans, it is possible that the perceptual organization of objects is analyzed early in visual processing as well. In fact, a lot of information about part structure would be obtained already in the process of segmenting an object into its parts—as, for example, indicated by studies on the early availability of symmetry information (e.g., [15,33,34]) and an extended neural network processing that information [35,36].

Previous studies developed concepts similar to our definition of part structure, all of which put emphasis on the information about relational aspects of object parts (e.g., [37,38,39,40,41,42]), even though some of these approaches do not discuss part segmentation as a perceptual problem. What many of these different concepts have in common is the reduction of visual information about object parts to a “neighborhood tree”, tree-like graph representations that mainly encode which parts are connected to which other parts. While this information alone might be sufficient to match many similar objects (e.g., [40]), thereby showing that even the most basic form of part structure is rich in information, this type of representation is not sufficient to explain our current findings. For example, a graph representation is insufficient to differentiate between the symmetrical and asymmetrical shape pairs of Experiment 1 where the connections between parts (as well as individual parts) are identical. Thus, an accurate account of the human visual representation of part structure has to include not only the connections between parts, but also relational information of unconnected parts (e.g., whether a limb has a mirror-symmetric counterpart on the other side of its parent). Medial axes representations (e.g., [14]) are a promising concept to encode these types of relational features (e.g., [43,44]).

Our results show that part structure is used to differentiate between superordinate classes, specifically those differing in their patterns of biological growth. It is highly feasible that we also represent categories of artificial objects with respect to their part structure: even though that structure might be less constrained, or constrained by alternative organizing principles (e.g., a helicopter is not the result of a growth process), most objects are composed in systematic ways according to particular laws that produce regularities in their part structure. It is also worth noting that in some cases there may even be similarities with classes of natural objects. For example, the part structure of an airplane is similar to that of a bird, given that both “solve” the problem of flight. This similarity in part structure between biological and artificial objects is further strengthened by the common constraints of structural integrity that only allow for certain viable arrangements. An exciting direction for future work is to identify other constraints, both universal and restricted to specific categories, and analyze how they affect the perceptual organization of objects, and consequently their mental representation and categorization.

## Figures and Tables

**Figure 1 brainsci-12-00667-f001:**
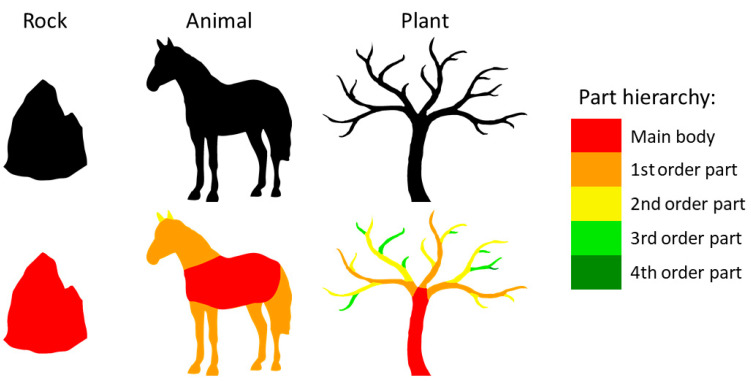
Scheme illustrating that different superordinate categories often show striking differences in their perceptual organization: rocks are usually made up of only one central body, without any limbs; animals tend to have prominent first-order parts growing from the main body with few higher-order parts; plants often have additional layers of parts further removed from the main body. Part organizations were created by the authors for illustrative purposes, not as an algorithm.

**Figure 2 brainsci-12-00667-f002:**
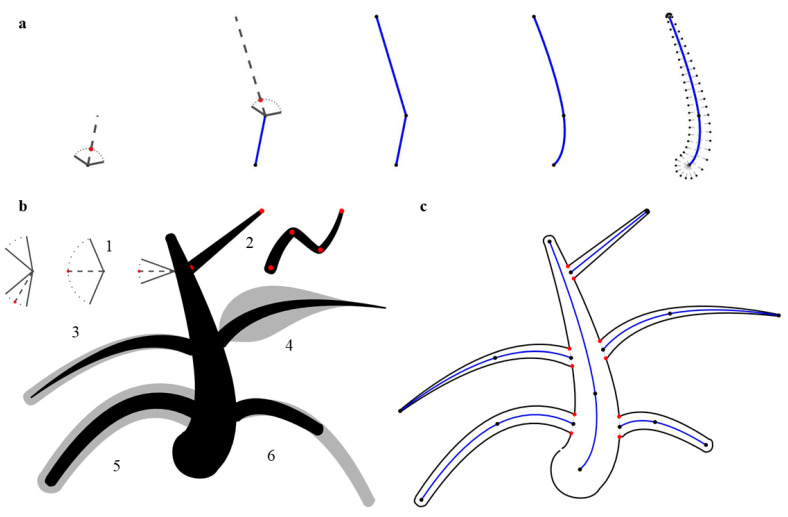
Overview of the shape creation algorithm. (**a**) Growth process of a part. From a starting point, we choose a random angle within a specified range (here, from −60° to +60° relative to a starting angle, which was pointed upwards for the first part, or orthogonal to the parent part in later parts). Then, we create a new point at a random distance deviating from the previous angle by a value chosen randomly from the same range as before. This process is repeated until the desired number of joints is created (the limbs of stimuli in this study ranged from three to five joints). The points are then connected by a line with each point defining a skeletal joint (panels 1–3). After smoothing the line (panel 4), we create silhouette points along the line with specified width values (panel 5). All parts are created analogously. (**b**) Examples of growth parameters. 1. The range of angles between joints can be varied from narrow to wide. 2. Number of joints. 3. Tapering. 4. For each joint, a width factor can be specified. Values between joints are interpolated. 5. Overall width relative to the parent. 6. Overall length relative to the parent. (**c**) Resulting skeletal representation. Joints (black dots) are shown together with skeletons of the main body and the parts (blue lines). Intersection points (red dots) between them are used for part segmentation.

**Figure 3 brainsci-12-00667-f003:**
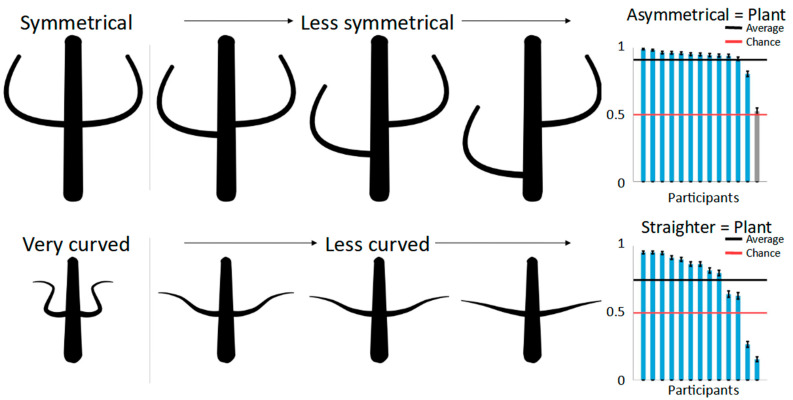
Stimuli and results of the 2-AFC task. The top row shows an object with symmetrical part pairs (left) and shapes becoming increasingly asymmetrical. Each shape pair in the 2-AFC task consisted of one symmetrical and one asymmetrical shape, and the participants were to choose the more plant-like shape. The results on a per-participant basis are plotted in the bar graph on the right with grey bars being not significantly different from chance (red line; evaluated by one-sided binomial test). The high average performance (black line) shows that asymmetry serves as a strong plant cue. The bottom row shows stimuli and findings for curvedness.

**Figure 4 brainsci-12-00667-f004:**
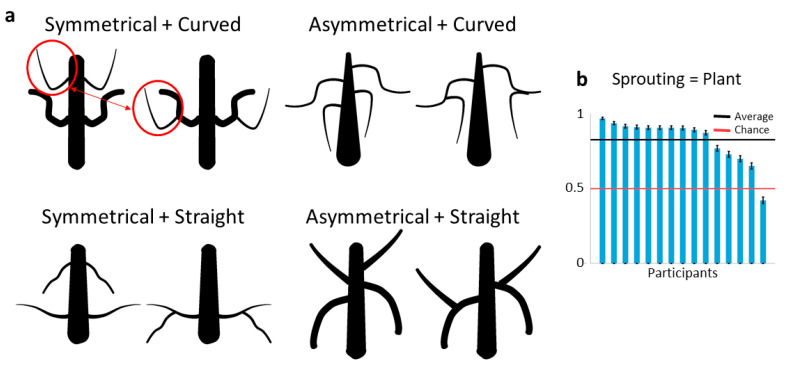
Sprouting limbs are a strong differentiation cue between plants and animals. (**a**) Types of stimuli created for the 2-AFC task combining symmetrical/asymmetrical and curved/straight limbs to make them appear more or less animal-/plant-like. (**b**) Results on a per-participant basis. All the participants’ responses were significantly different from chance (red line; evaluated by one-sided binomial test). The participants were instructed to pick the more plant-like shape versus the more animal-like shape.

**Figure 5 brainsci-12-00667-f005:**
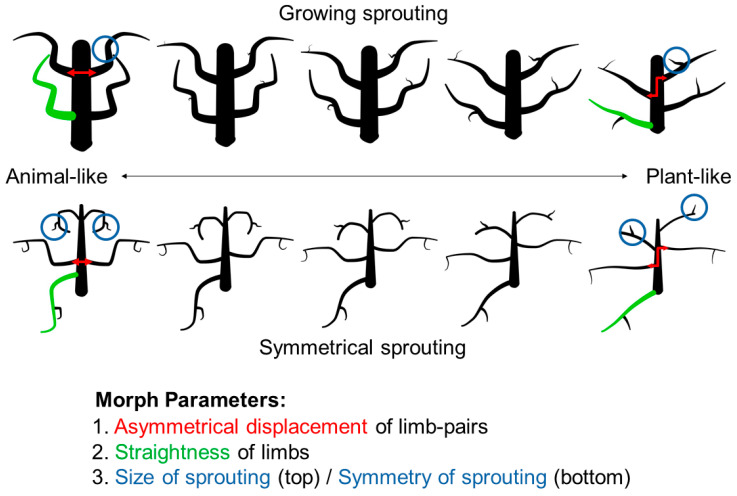
Morphing sequences from animal- to plant-like and the corresponding parameters. The top row shows an example sequence for the “growing sprouting” condition and the bottom row for the “symmetrical sprouting” condition. Changes in three morph parameters are illustrated: symmetrical limb pairs are increasingly displaced (red), parts get continuously straighter (green), and sprouting parts (blue) either grow (from non-existent to largest) from random starting points (top row of shapes) or are always the same size but make up symmetrical pairs where possible (bottom row).

**Figure 6 brainsci-12-00667-f006:**
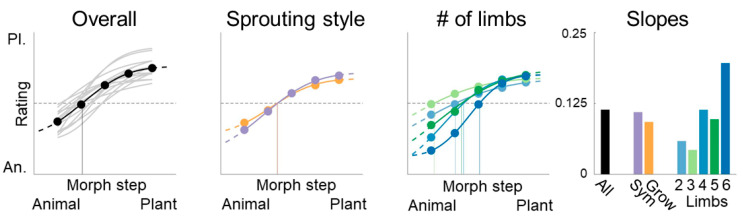
Morph sequences agree with human categorizations. Panel 1, aggregated responses (y-axis, Pl. = plant, An. = Animal) as a function of morphing steps (x-axis), with each participant’s average response in faint grey, shows the overall agreement between the morphing steps and the observers’ categorizations. The midpoint of the animal/plant spectrum used by observers is indicated by a dashed line. Panel 2, different sprouting styles (growing sprouting in orange and symmetrical sprouting in purple) had very similar effects on categorization. Panel 3, the more parts a shape had, the stronger was the difference between the morphing steps. Panel 4, slopes of curves in Panels 1–3. Noticeably, the parts with an odd number of non-main body limbs have a smaller slope and are more biased towards plant-likeness compared to the shapes with an even number of limbs.

## Data Availability

All the stimuli will be made available on Zenodo upon acceptance.

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
