# Peer review of "Superordinate Categorization Based on the Perceptual Organization of Parts"

_brainsci, 2022, doi:10.3390/brainsci12050667_

Round 1
Reviewer 2 Report
The paper is well written and clear. The results are interesting. Based on a forced-choice task (plant or animal) we see that novel shapes tend to be classified as animal if they are more symmetrical and smoothly curved, and fewer sprouting parts.
I only have a few suggestions. It is important to say a bit more about the task for the participants, they were asked to decide between part or animal, but what where they told this was about? These are simple shapes, not likely to be seen as real plants or animals. Where people ask to judge how much these shapes matched actual plant/animals or whether they would look good if used as representations of plant/animals?
With respect to sprouting, a qualification may be necessary. Symmetry of the sprouting was manipulated, in terms of the overall figure, but not the local symmetry of the parts. If we think of the fingers of the hand, they are sort of sprouting, but they are also quite similar to each other. In other words, the fact that the sprouting created independent shapes (different in size and shape) they may be best perceived as branches, rather than limbs.
with respect to references, there is a need for some tiding up. Some journals are abbreviated and not others, some have DOI and not others, and in this case the publisher is missing: Stevens, P.S. Patterns in Nature; 1974
